# Sizing of a Plug-In Hybrid Electric Vehicle with the Hybrid Energy Storage System

**Jian Tu \*, Zhifeng Bai and Xiaolan Wu**

Electromechanic Engineering Institute, Xi'an University of Architecture and Technology, Xi'an 710311, China; zhifeng.bai@xauat.edu.cn (Z.B.); wuxiaolan@xauat.edu.cn (X.W.)
\* Correspondence: tujian@xauat.edu.cn; Tel.: +86-135-7298-0720

**Abstract:** For plug-in hybrid electric vehicle (PHEV), using a hybrid energy storage system (HESS) instead of a single battery system can prolong the battery life and reduce the vehicle cost. To develop a PHEV with HESS, it is a key link to obtain the optimal size of the power supply and energy system that can meet the load requirements of a driving cycle. Since little effort has been dedicated to simultaneously performing the component sizing of PHEV and HESS, this paper proposes an approach based on the particle swarm optimization (PSO) algorithm to simultaneously determine the sizes of the engine, motor, battery and supercapacitor (SC) in a PHEV with HESS. The drivetrain cost is minimized in a different all-electric range (AER)—and depends on the battery type—while ensuring the driving performance requirements. In addition, the effects of the power system and drive cycle on the component sizes were analyzed and compared. The simulation results show that the cost of the PHEV drivetrain with the Ni-MH battery/SC HESS is reduced by up to 12.21% when compared to the drivetrain with the Li-ion battery/SC HESS. The drivetrain cost is reduced by 8.79% when compared to analysis-based optimization. The type of power supply system and drive cycle can significantly affect the optimization results.

**Keywords:** plug-in hybrid electric vehicle; hybrid energy storage system; particle swarm optimization; component sizing; all electric range; cost





## 1. Introduction

Nowadays, plug-in hybrid electric vehicles (PHEVs) are attracting increasing attention from the automotive industry [1]. Compared with traditional hybrid electric vehicles, PHEVs are equipped with larger capacity batteries that can be charged from the power grid, which greatly reduce the energy consumption cost and carbon dioxide emissions [2]. However, PHEVs often encounter significant instantaneous power demand during the driving process, which leads to batteries being frequently charged and discharged, thus accelerating their aging [3–5]. A supercapacitor has a higher power density and can withstand high current, thus the hybrid energy storage system (HESS) composed of batteries and supercapacitors greatly reduces the peak power of the battery and prolongs the battery life. In addition, it also has the advantages of high energy utilization and high safety [6,7]. Therefore, to meet the requirements of PHEV for the high energy and high power density of on-board energy storage system, using a hybrid energy storage system as energy source is an economic and safe solution.

To develop a PHEV with HESS, it is a key link to obtain the optimal size of power supply and energy system that can meet the load requirements of the driving cycle [8]. In the literature, most studies have focused on sizing the components of PHEV with a single battery system, which can be classified into analysis-based and optimization-based methodologies. Yang used the parameter analysis model of vehicle energy consumption and a rapid dynamic programming to analyze the component size [9]. Tran obtained the component sizing of PHEV through analysis according to the requirements of dynamic

performance [10]. Guo combined a multi-island genetic algorithm and a dynamic programming algorithm to determine the optimum speed ratio for the redesigned driveline [11]. Madanipour used the genetic algorithm to optimize the size of the engine, motor and battery, which take the minimum weighted fuel consumption and weighted exhaust emission as the optimization objective and the driving performance requirements as the constraint. The results show that the fuel consumption is reduced by 27% on average compared with that before optimization [12]. Vahid proposed a multi-objective optimization algorithm for the component sizing optimization of a PHEV, wherein the objective function was defined to minimize the drivetrain cost and exhaust emissions. The final result shows that the operating cost is reduced by up to 10%, and the exhaust emissions is reduced by up to 17% [13]. Wu proposed a convex optimization to minimize the energy cost and power sources cost. The results show that the optimal battery rated power is 54 kw and the energy capacity is 29 kwh [14]. Nikolce proposed a novel convex modeling approach to optimize the battery size and energy management strategy of a PHEV at the same time. The final result shows that the operating cost is reduced by up to 8.46% when comparing with results obtained by dynamic program [15]. Mitra proposed a convex optimization to minimize a weighted sum of the component and operational costs. The results show that the cost of a PHEV50 is 14% higher than the best design [16].

As for the HESS, the component sizes are usually decided by optimization under a certain power source. Bai used a dynamic programming algorithm to minimize the fuel economy and battery life degradation rate. These results show that the battery aging rate is reduced by 48.9% [17]. Song used a two-dimensional PMP to determine that size of the components and an EMS design for the HESS in a PHEV, wherein the optimization objective was that of the operating cost, including the fuel cost and the electricity cost. The simulation results show that the operation cost is reduced by 28.6% compared with the traditional hybrid system without a supercapacitor [18]. Hong used a two-dimensional PMP to achieve the optimal sizing of the powertrain by minimizing the energy consumption and system degradation in a PHEV [19]. Zhang used a genetic algorithm (GA) to minimize the HESS initial cost, setting the vehicle power performance requirements as the constraints. The results show that the cost is lower than the analysis-based optimization [20]. Omar used a PSO to achieve the optimal sizing of the powertrain by minimizing the cost, volume and mass of the fuel cell and the supercapacitor in a fuel cell hybrid electric vehicle (FCHV) [21].

As the literature review reveals, little attempt has been dedicated to simultaneously perform the component sizing of PHEV and HESS. Therefore, this paper proposes an approach based on the PSO algorithm to simultaneously determine the sizes of the engine, motor, battery and supercapacitor in a PHEV with HESS. The objective function is defined to minimize the drivetrain cost including the initial cost and replacement cost, while setting the driving performance requirements as the constraints. Considering different customer requirements, component sizing is carried out for two types of batteries and three different all-electric ranges (AERs). In addition, the effects of the power system and drive cycle on component sizes are analyzed and compared. By performing the PSO for different optimization variable candidates, the optimal sizes of the engine, motor, battery and supercapacitor are globally found. The advanced vehicle simulator (ADVISOR) and MATLAB are used to investigate the effectiveness of the proposed approach. Simulation results show that the drivetrain cost of the Ni-MH battery/SC HESS battery is reduced by more than 10% when compared to the Li-ion battery/SC HESS; and the PSO is effective in reducing the drivetrain cost compared with analysis-based optimization. It is effective to reduce the drivetrain cost by adding the SC to PHEV. The driving cycle aggressiveness can significantly affect the optimization results.

The paper is organized as follows. In Section 2, the configuration and dynamic model of the PHEV with HESS is illustrated. Section 3 proposes an efficient component sizing optimization methodology. The optimization results are shown in Section IV. Finally, Section V gives the conclusion.

## 2. Modeling and Control Strategy

### 2.1. Vehicle Configuration

This paper used a parallel PHEV with a HESS, and the battery and the supercapacitor are connected in parallel, whilst the DC/DC converter and inverter provide power for the motor [22,23]. Internal combustion (IC) engine and motor may provide power to the vehicle wheels. The structure is shown in Figure 1 and the vehicle parameters are shown in Table 1.

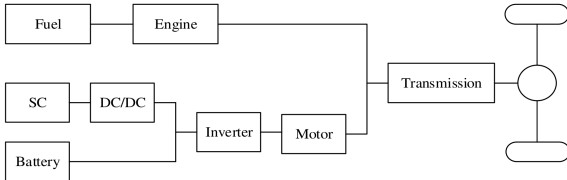

**Figure 1.** Topology of a parallel PHEV with an HESS.

**Table 1.** Base vehicle parameters.

| Parameter | Value |
| --- | --- |
| Glider mass (kg) | 1150 |
| Rolling resistance coefficient | 0.009 |
| Air drag coefficient | 0.3 |
| Frontal area (m$^2$) | 2.17 |
| Engine power (kW) | 41 |
| Motor power (kW) | 58 |
| Li-ion battery capacity (Ah/module) | 30 |
| Li-ion battery voltage (V/module) | 12 |
| Ni-MH battery capacity (Ah/module) | 28 |
| Ni-MH battery capacity (Ah/module) | 6 |
| Supercapacitors capacity (F/module) | 3000 |
| Supercapacitors voltage (V/module) | 2.5 |

### 2.2. Model

ADVISOR is an advanced vehicle simulation software developed by NREL, a renewable energy laboratory in the United States. In this paper, the model of HESS and the control strategy are established in Simulink and added to the advisor model to obtain a PHEV's model with HESS [24].

#### 2.2.1. Battery Model

Figure 2 shows the battery model. $U_b$ is the open circuit voltage (OCV), $i_b$ is the current and $R_b$ represents inner resistance. $SOC_b$ represents the remaining capacitor of the battery in the current state:

$$\begin{cases} SOC_b = SOC_{b,0} - \frac{1}{Q_b} \int i_b(t)dt \\ SOC_{b,\min} \leq SOC_b \leq SOC_{b,\max} \end{cases} \tag{1}$$

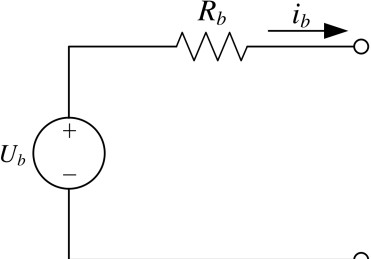

**Figure 2.** Battery model.

### 2.2.2. Supercapacitor Model

Figure 3 shows the supercapacitor model. $C_{uc}$ and $R_{uc}$ represent the supercapacitor capacity and inner resistance, respectively. $U_{uc}$ is the supercapacitor OCV, $I_{uc}$ is the current and $P_{uc}$ is the supercapacitor output power.

$$\begin{cases} u_{uc} = \frac{1}{c_{uc}} i_{uc} \\ u_{uc,\min} \le u_{uc} \le u_{uc,\max} \end{cases} \tag{2}$$

$$P_{uc} = \left( i_{uc} i_{uc} - i_{uc}^2 R_{uc} \right) n_{uc} \tag{3}$$

$$i_{uc} = \frac{u_{uc} - \sqrt{u_{uc}^2 - 4 P_{uc} R_{uc} / n_{uc}}}{2 R_{uc}} \tag{4}$$

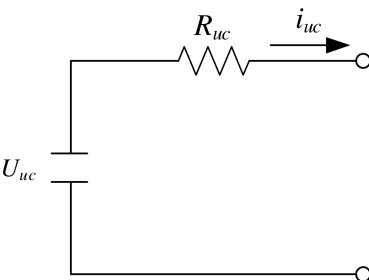

**Figure 3.** Supercapacitor model.

### 2.3. Hybrid Drive Control Strategy

This paper use the charge depleting (CD) charge sustaining (CS) control strategy to determine the energy distribution of the engine and motor. As shown in Figure 4a, when the battery is fully charged, PHEV operates in CD mode until the electric energy drops below the target value (Sobj), and the vehicle turns to CS mode.

Figure 4b shows the CS mode (flag = 0) if the required torque is lower than the critical torque of the engine, and the motor drives the vehicle alone. If the required torque is greater than the maximum torque of the engine, the vehicle is driven by the engine and motor. At a given torque and speed, if the engine efficiency is low, the full torque is provided by the motor.

Figure 4c shows the CS mode (flag = 1): if the required torque is excessively small, the engine works on its minimum torque curve, and the additional torque drives the motor to charge the battery. In other cases, the output of the engine is the required torque of the whole vehicle plus the charging torque.

### 2.4. Hybrid Energy Storage System Control Strategy

This paper uses a logic threshold control strategy to determine the battery and supercapacitor's energy distribution, as shown in the flow chart in Figure 5, where $P_{req}$ is the power demand of the motor on the HESS; $P_{bat}$ is the demand power of the battery; $P_{cap}$ is the demand power of the supercapacitor; and $F_1(s)$ is the filter function. The control rules are as follows.

In driving mode, when the *SOC* of the supercapacitor (*SOC*2) is lower than the lower limit (*SOC*2$_{\min}$), only the battery supplies power; when the *SOC* of the supercapacitor is in a normal state and the required power of the motor is less than the threshold value ($P_p$), the battery provides power for the vehicle, and when the required power of the motor exceeds Pp, the remaining power is supplied by the supercapacitor.

In braking mode, when the *SOC*2 does not reach the maximum (*SOC*2$_{\max}$) and the braking power does not reach the threshold value ($P_n$), the supercapacitor is charged, and when the braking power exceeds Pn, the battery and supercapacitor are charged at the same time; when the *SOC*2 reaches its maximum, the battery recovers energy alone.

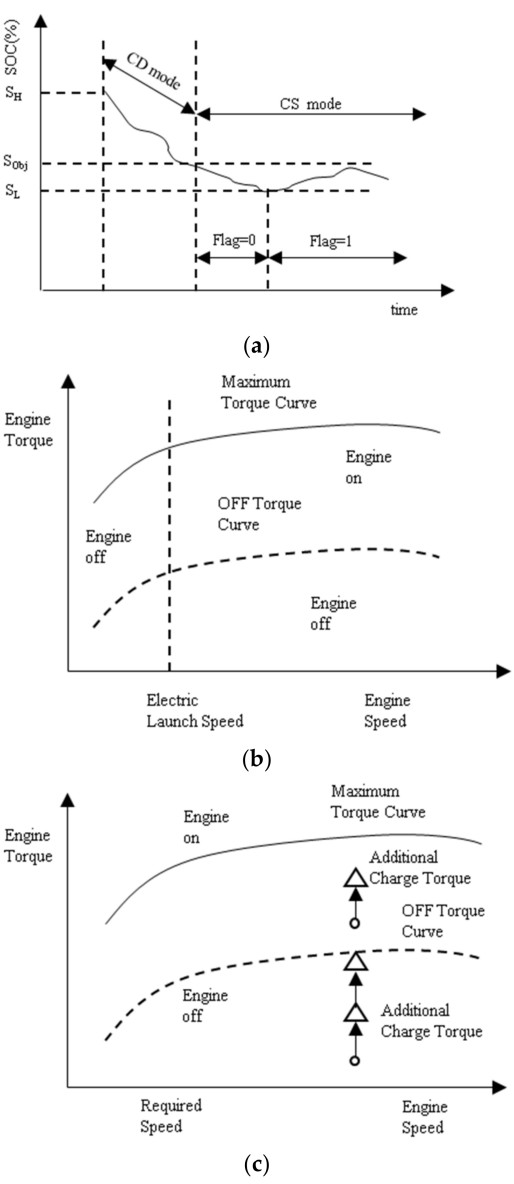

**Figure 4.** Hybrid drive control strategy. (**a**) CDCS strategy; (**b**) CS mode (flag = 0); and (**c**) CS mode (flag = 1).

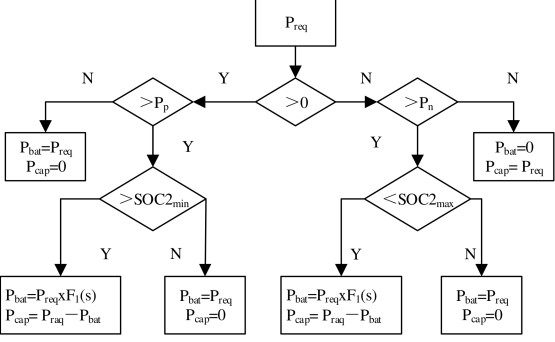

**Figure 5.** HESS control strategy.

## 3. PHEV Component Sizing Optimization

In this paper, the torque scaling factors of the engine ($S_{IC}$), the torque scaling factors of the motor ($S_{EM}$), the battery modules number ($N_B$) and the capacity scaling factor ($S_{BC}$),

supercapacitor modules number ($N_{UC}$) and the capacity scaling factor ($S_{UC}$) are varied during optimization. The boundary values were estimated from the theoretical analysis. The parameters are shown in Table 2.

**Table 2.** Upper and lower bounds of optimization variables for two types of battery.

| Optimization Variable | Lower Bound | Upper Bound |
|---|---|---|
| $S_{IC}$ | 0.8 | 1.8 |
| $S_{EM}$ | 0.6 | 1.5 |
| $N_B$ | 25 | 65 |
| $N_{UC}$ | 65 | 120 |
| $S_{BC}$ (AER40 km) | 0.7 | 1.4 |
| (AER60 km) | 1.0 | 2.0 |
| (AER80 km) | 1.4 | 2.7 |
| $S_{BC}$ (AER40 km) | 0.8 | 1.8 |
| (AER60 km) | 1.1 | 2.1 |
| (AER60 km) | 1.4 | 2.5 |
| $S_{UC}$ (AER40 km) | 0.3 | 1.1 |
| (AER60 km) | 0.4 | 1.2 |
| (AER60 km) | 0.5 | 1.3 |

In order to ensure the driving performance requirements, in reference to the performance of the prototype vehicle and the national standard GB/T 19752-2005, the conditions that the whole vehicle should meet are shown in Table 3.

**Table 3.** Vehicle performance constraints.

| Constraints | Description |
|---|---|
| Acceleration | 0–97 km/h (0–60 mph) $\leq$ 12 s |
| time | 64–97 km/h (40–60 mph) $\leq$ 5.3 s |
| | 0–137 km/h (0–85 mph) $\leq$ 23.4 s |
| | 0–48.3 km/h (0–30 mph) $\leq$ 5 s |
| | in motor alone |
| Gradeability | >30% (15 km/h) |
| Maximum speed | $\geq$170 km/h |

The objective function is defined to minimize the drivetrain cost, including the cost of the engine, motor and HESS. The cost of HESS is composed of the initial cost and the battery replacement cost. Because the cycle life of the supercapacitor group is very long, it does not need to be replaced during the warranty period, and thus only the replacement cost of battery group is calculated [25]:

$$C_{total} = C_E + C_M + C_{bat\_init} + C_{uc\_init} + C_{bat\_rep} \tag{5}$$

where $C_E$ is the engine cost, $C_M$ is the motor cost, $C_{bat\_init}$ is the initial cost of the battery, $C_{uc\_init}$ is the cost of the ultracapacitor, and $C_{bat\_rep}$ is the replacement cost of the battery, which can be expressed as follows [26,27]:

$$C_E = 77.34 P_E + 2735 \tag{6}$$

$$C_M = 140.34 P_M + 2739 \tag{7}$$

$$C_{bat\_init} = c_{bat} C_{bat} U_{bat} \tag{8}$$

$$C_{uc\_init} = c_{uc} \frac{0.5 C_{uc} \left( V_{uc-max}^2 - V_{uc-min}^2 \right) N_{uc}}{1000} \tag{9}$$

$$C_{bat\_rep} = n_r c_{bat} C_{bat} U_{bat} \tag{10}$$

where $P_E$ is the engine peak power in kW, $P_M$ is the motor peak power in kW, $C_{bat}$ is the battery cost, $C_{uc}$ is the cost of supercapacitors, and $n_r$ is the number of batteries' replacement times [28].

According to the test methods in the national standard GB/T 31484—2015, the replacement condition of batteries is that the actual capacity is reduced to 80% of the rated capacity. According to the Standard for Compulsory Scrapping of Motor Vehicles, when the cumulative mileage of the vehicle reaches 600,000 km, the vehicle will be scrapped. The number of batteries' replacement times within the life span of the vehicle is thus as follows:

$$n_{\mathbf{r}} = \frac{600000 A_{day}}{L_{day} A_{life}} \tag{11}$$

where $A_{day}$ is the daily average ampere–hour circulation of the battery pack in Ah, including the cycling and charging conditions [29]; $A_{life}$ is the cycling life of the battery in Ah [30,31]; and $L_{day}$ is the average daily mileage in km.

In summary, the optimization mode is:

$$\begin{aligned} x &= [S_{IC}, S_{EM}, N_B, S_{BC}, N_{UC}, S_{UC}] \\ &miny = F(x) \\ &F(x) = C_{total} \\ subjuct\ to\ &a_i \le x_i \le b_i, i = 1, 2 \cdots, n \\ &hj(x) \le 0 = 1, 2, \ldots, p \\ &g_k(x) \le 0 k = 1, 2, \cdots, l \end{aligned} \tag{12}$$

where $a$ and $b$ are the lower and upper bounds of the optimization variable, $h_j\ (x)$ and $g_k\ (x)$ are p-dimensional equality constraints and one-dimensional inequality constraints, respectively. $h_j\ (x)$ and $g_k\ (x)$ determine the feasible range of decision variables jointly.

*PSO Solution*

Firstly, the algorithm assigns velocities and initial random positions' velocities to all particles in the space, the best particle of the personal (pbest), and the best particle of the swarm (gbest) to advance the position of each particle in turn [32]. The equation is described as follows:

$$v_{i+1} = \omega v_i + c_1 r_1 ( pbest_i - x_i) + c_2 r_2 (gbest_i - x_i) \tag{13}$$

$$x_{i+1} = v_{i+1} + x_i \tag{14}$$

where $c_1$ is the cognitive parameter, $c_2$ is the social parameter, $c_1 = 0.5$, $c_2 = 2.0$. $r_1$ and $r_2$ are random numbers, the range is [0, 1] and it is uniformly distributed. $\omega$ is the inertia weight, $\omega = 0.8$. Equation (13) provides the $i$-th particle's new velocity. The new position of the $i$-th particle is determined by Equation (14) at each iteration. The particle will be iteratively updated using these formulas until an optimal solution is obtained or the number of iterations is reached. The optimization flow is shown in Figure 6:

Step 1: Initialize the vehicle model and swarm, and the swarm must be in the optimization interval.

Step 2: Assign the individual values in the population to model in turn for simulation, obtaining the vehicle performance and judging whether the constraint condition is met, if not, eliminating the individual value; but if so, outputting the total cost, as well as updating the pbest, gbest and the speed and position of each particle.

Step 3: Judging whether the end of condition is met, if not, carrying out iterative optimization, calculating the speed and position of each particle of a new population, and repeating Step 2; if so, outputting an optimal optimization result.

The number of the population of PSO is set to 20, and the number of iterations is set to 20. In order to simulate the constraint of AER, the motor drives the vehicle alone. The initial *SOC* of the battery is set to 0.9, and the minimum *SOC* is set to 0.2. The Urban Dynamometer

Driving Schedule (UDDS) is selected as the target driving cycle, and 40 km, 60 km and 80 km are selected as the different AERs. During the simulation, if the incremental trajectory exceeds the specified value (3.2 km/h), the AER constraint is not satisfied.

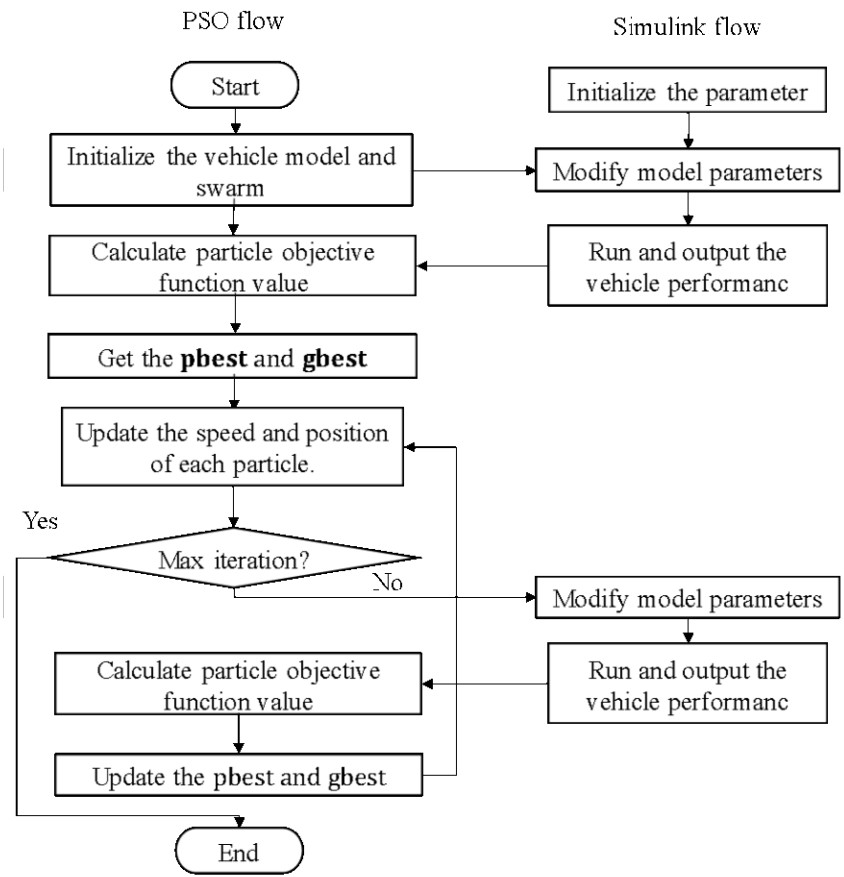

**Figure 6.** Flow chart of optimization.

## 4. Optimization Results

Table 4 summarizes the best parameters obtained based on the UDDS driving cycle and the two types of batteries. The final component sizes can be determined by these parameters. In addition, the vehicle mass is the sum of the base vehicle mass and the mass of the optimized components, including the motor, engine, battery and supercapacitor. This is shown in Table 5.

**Table 4.** Optimum PHEV specifications for two types of battery.

| Battery Type | AER (km) | $S_{IC}$ | $S_{EM}$ | $N_B$ | $S_{BC}$ | $N_{UC}$ | $S_{UC}$ |
|---|---|---|---|---|---|---|---|
| Li-ion | 40 | 0.9976 | 0.7310 | 32.6252 | 0.9207 | 115.7019 | 0.5414 |
| Ni-MH | 40 | 1.0975 | 0.8068 | 49.1521 | 1.3856 | 108.5634 | 0.6382 |
| Li-ion | 60 | 1.0463 | 0.8621 | 29.7014 | 1.5304 | 110.2505 | 0.6495 |
| Ni-MH | 60 | 1.1926 | 0.9155 | 49.0465 | 2.0753 | 108.0355 | 0.6995 |
| Li-ion | 80 | 1.1293 | 0.9052 | 35.0756 | 1.8425 | 102.3658 | 0.7492 |
| Ni-MH | 80 | 1.3121 | 0.9810 | 60.8873 | 2.3406 | 108.6984 | 0.8256 |

**Table 5.** Optimal specifications for two types of battery based on the UDDS.

| Battery Type | AER (km) | Mass (kg) | Engine Power (kW) | Motor Power (kW) | Battery Capacity (Ah) | Battery Energy (kWh) | Supercapacitor Energy (Wh) | Drivetrain Cost (CNY) |
|---|---|---|---|---|---|---|---|---|
| Li-ion | 40 | 1609 | 40.9 | 42.4 | 27.6 | 10.94 | 351 | 118,013 |
| Ni-MH | 40 | 1701 | 45.0 | 46.8 | 38.8 | 11.44 | 386 | 101,525 |
| Li-ion | 60 | 1680 | 42.9 | 50.0 | 45.9 | 16.53 | 401 | 141,862 |
| Ni-MH | 60 | 1804 | 48.9 | 53.1 | 58.1 | 17.08 | 415 | 126,163 |
| Li-ion | 80 | 1752 | 46.3 | 52.5 | 55.3 | 23.21 | 432 | 177,401 |
| Ni-MH | 80 | 1935 | 53.8 | 56.9 | 65.5 | 23.94 | 499 | 156,829 |

As shown in Figure 7, different AERs and different battery types can significantly affect the optimal design variables. As the AER increases, so does the battery power and vehicle mass. Meanwhile, the engine and motor need to provide more energy to drive the vehicle and meet the driving performance requirements, their sizes also increase. Thus, the drivetrain cost is increased. In addition, for the same AER, because a Li-ion battery has greater energy density, its total mass is lower than that of the Ni-MH battery, and thus the battery energy, engine, and motor parameters are lower. However, the high cost of an Li-ion battery caused a higher drivetrain cost. The cost of an PHEV drivetrain with an Ni-MH battery/SC HESS is reduced by up to 12.21% when compared to a drivetrain with Li-ion battery/SC HESS.

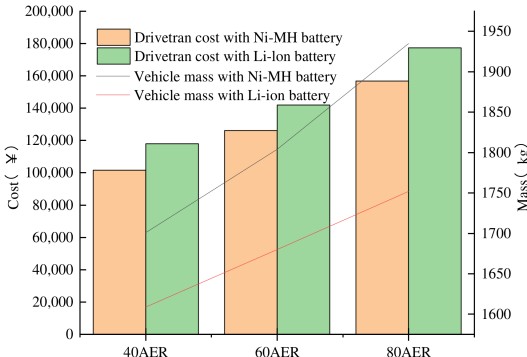

**Figure 7.** Drivetrain cost and vehicle mass for different AER and battery.

In order to evaluate the superiority of the PSO, the results using the Li-ion battery/SC are compared with the results of an optimization method based on theoretical analysis. The method has the following characteristics. Firstly, obtain the maximum power and economic power of the engine according to the speed performance constraint. Secondly, the maximum required power of the vehicle is obtained according to the acceleration and gradeability performance constraint. The motor power is equal to the maximum required power of the vehicle minus the engine economic power. Lastly, the battery and supercapacitor energy are obtained according to the AER and motor-only acceleration performance constraint. The PHEVs designed by the Li-ion battery optimization method and based on theoretical analysis are shown in Table 6. Figure 8 compares the powertrain costs obtained from the two described methods. As shown in this Figure, the drivetrain cost by this method is reduced by 8.79%.

**Table 6.** Optimum specifications for Li-ion battery type based on theoretical analysis.

| AER | Engine (kW) | Motor Energy (kW) | Battery Energy (kWh) | Supercapacitor Energy (Wh) | Mass (kg) | Drivetrain Cost (CNY) |
|---|---|---|---|---|---|---|
| 40 | 54.2 | 57.0 | 11.26 | 362 | 1671 | 123,706 |
| 60 | 54.7 | 61.8 | 17.21 | 429 | 1742 | 161,792 |
| 80 | 55.1 | 64.6 | 23.62 | 458 | 1802 | 193,958 |

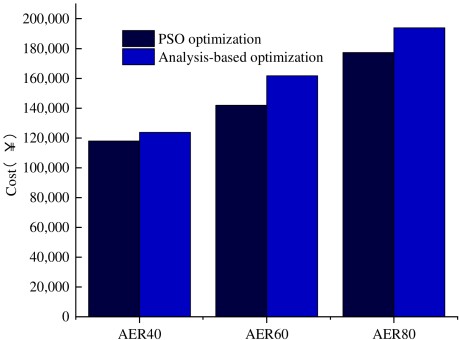

**Figure 8.** Drivetrain cost for Li-ion battery type with different approach.

### 4.1. Effect of Supercapacitors on Component Sizing

In order to investigate the effect of supercapacitors on component sizing, a Li-ion battery system is selected for optimization, and the optimized component sizes are compared in Figure 9. The obtained vehicle mass and drivetrain costs are compared in Figure 10. In these two figures, P1 represents the hybrid energy storage system and P2 represents the Li-ion battery system. The parameters of the engine and motor are slightly reduced compared to an HESS, which is due to the reduction in the total vehicle mass. On the other hand, because the reduction in the power performance of the Li-ion battery system, it is difficult to meet the high power demand when the *SOC* of the battery is low, so more battery energy is needed. Because the supercapacitor is expensive, the initial cost of an Li-ion battery system is greatly reduced, but the supercapacitor in the HESS can withstand the instantaneous high power, and thus the battery current is greatly reduced. As shown in Figure 11, the maximum discharge current is reduced from 45.25 A to 21.60 A. The maximum charging current is reduced from −34.12 A to −10.85 A, and the drivetrain cost is reduced by up to 12.34%, which proves that the supercapacitors can extend the battery life and reduce the battery replacement cost.

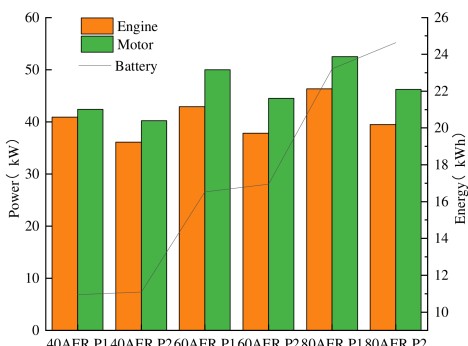

**Figure 9.** Optimum component sizes for different energy storage systems.

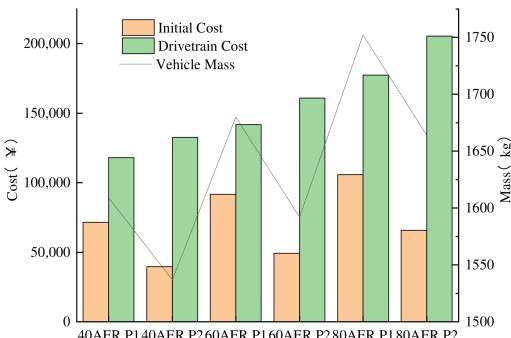

**Figure 10.** Drivetrain cost and vehicle mass for different energy storage systems.

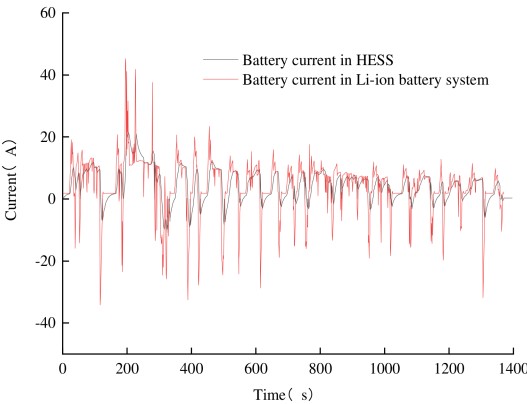

**Figure 11.** Battery current for different energy storage systems.

### 4.2. Effect of Driving Cycle on Component Sizing

In order to study the effect of a drive cycle on the component sizing, the optimization problems of the Highway Fuel Economy Test (HWFET), LA92 (Los Angeles 92) and US06 (high speed and high acceleration component of SFTP) driving cycle are solved, respectively. The optimization results for the Li-ion battery, 60AER and common driving performance requirements are compared in Figures 12 and 13. The vehicle mass and drivetrain costs are compared in Figure 14. As shown in these figures, the energy of battery and supercapacitor obviously increases with the cycle aggressiveness, and thus the parameters of engine and the motor also increase, and the vehicle mass d and drivetrain cost are higher.

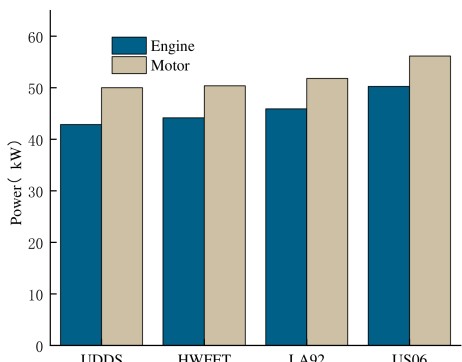

**Figure 12.** Engine and motor sizes for different driving cycles.

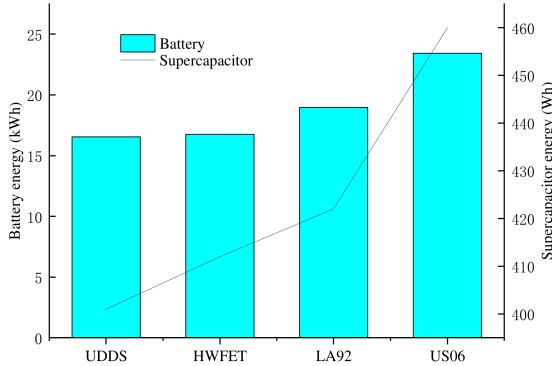

**Figure 13.** Battery and supercapacitor sizes for different driving cycles.

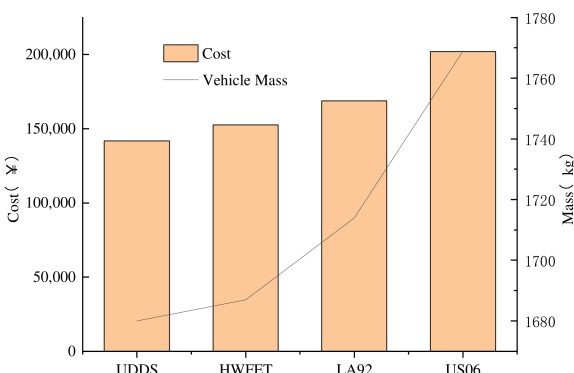

**Figure 14.** Drivetrain cost and vehicle mass for different driving cycles.

## 5. Conclusions

In this paper, a PSO algorithm was used to determine the sizes of the components of a PHEV with a HESS. The parameters of the engine, motor, battery and supercapacitor were selected as optimization variables, The drivetrain cost was minimized while ensuring the driving performance requirements. Three kinds of AER and two types of batteries are optimized. The optimization results show that:

(1) The drivetrain cost of an HESS with a Ni-MH battery is reduced by up to 12.21% when compared to a HESS with Li-ion battery. Compared to the results from theoretical analysis, the drivetrain cost optimized by PSO is reduced by 8.79%.

(2) After adding the supercapacitor to the energy storage system, the parameters of the engine and motor slightly increased, and the initial cost is higher, but the supercapacitor can extend the battery life and thus the drivetrain cost is reduced by 12.34% compared to an energy storage system without supercapacitors.

(3) In order to study the effect of a drive cycle on component sizing, choose three different drive cycles to optimize. The simulation results show that the parameters of the engine, motor, battery and supercapacitor are increased with the cycle aggressiveness, and the vehicle mass and drivetrain cost are higher.

In order to improve the efficiency of PHEV as much as possible, a reasonable energy management strategy is needed. Future research will focus on the simultaneous selection of component sizes and control parameters.

**Author Contributions:** Conceptualization, J.T., Z.B. and X.W.; data curation, J.T.; formal analysis, J.T.; investigation, J.T.; methodology, J.T., Z.B. and X.W.; software, J.T., Z.B. and X.W.; supervision, Z.B. and X.W.; validation, J.T.; visualization, J.T.; writing—original draft preparation, J.T.; writing— review and editing, J.T., Z.B. and X.W. All authors have read and agreed to the published version of the manuscript.

**Funding:** Applied Technology R & D Projects of Beilin District in 2020, GX2012; Natural Science Basic Research Program of Shaanxi, 2021JM-363.

**Institutional Review Board Statement:** Not applicable.

**Informed Consent Statement:** Not applicable.

**Data Availability Statement:** The data that support the findings of this study are available from the corresponding author upon reasonable request.

**Conflicts of Interest:** The authors declare no conflict of interest.

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
