# Peer review of "Sizing of a Plug-In Hybrid Electric Vehicle with the Hybrid Energy Storage System"

_wevj, doi:10.3390/wevj13070110_

Round 1
Reviewer 1 Report
Dear Authors,
Your work seems interesting and is one of many research and is one of the many works to investigate the optimization of the size of a hybrid energy storage system. Some points should be improved/corrected:
1. Correct errors in sentences and typos - check the text again, because it contains many small typos (also in the pictures, e.g. fig.1), measurement units (e.g. kw instead of kW - fig. 8,12,13) ​​are incorrectly written, often inconsistent with the text (no lower indexation, Nb, Nu in Table 1), errors in sentences (e.g. lines 19, 76-78, 276-282, 299, 303, 313 and others).
2. Explain all the abbreviations used for the first time, also in the figures (fig.5 no explanation of the abbreviations used Prep, Pcap), UDDS (line 213), s.t.ai (eq.12).
3. Subsection 2.2. - I suggest adding a link to the website or the literature on the software used.
4. Tab.2. - I suggest changing the table layout so that it is clear what the individual values ​​represent. As it stands, you have to guess which values ​​in the last column are the lower or upper limits of the optimization.
Where do the presented boundary values ​​come from? On what basis were they estimated?
5. Equations 6 and 7 - where do the presented numerical values ​​come from? On what basis were they estimated?
6. What is the novelty of your work? Please highlight, especially in the abstract.
Author Response
Dear reviewer,
Thank you very much for the helpful review. In the appendix, you can find our answers to your questions.

Reviewer 2 Report
The submitted manuscript titled as “Sizing of a plug-in hybrid electric vehicle with the hybrid energy storage system” is a good application. However, following are the suggestions that are needed to be incorporate.
1. The authors should discuss the more articles on other previously used metaheuristics algorithms for the particular problem.
2. The key contributions of the proposed study after the end of introduction part are missing.
3. Increase the font size of figure 7-14. In the current form it is hard to analyse the figures.
4. Mention the conclusion point wise.
Author Response

(The authors gave the same response as above.)

Round 2
Reviewer 1 Report
Dear Authors,
Thank you for making corrections, but please use the correct power unit abbreviation [kW - kilowatts] - e.g. lines 192, tables 5 and 6 and check again your work.
Author Response
Dear reviewer,
Thank you very much for your valuable comments. We have corrected the wrong abbreviation and checked again our work.( e.g. lines 192,
tables 1, 5 and 6, fig9, 12 and 13). So we hope it can meet the journal’s standard.
